# Vitamin B_12_ Auxotrophy in Isolates from the Deep Subsurface of the Iberian Pyrite Belt

**DOI:** 10.3390/genes14071339

**Published:** 2023-06-26

**Authors:** Guillermo Mateos, Adrián Martínez-Bonilla, José M. Martínez, Ricardo Amils

**Affiliations:** 1Centro de Biología Molecular Severo Ochoa (CBMSO), Calle Nicolás Cabrera 1, 28049 Madrid, Spain; gmateos@cbm.csic.es (G.M.); amartinez@cbm.csic.es (A.M.-B.); jmm.lozano@cbm.csic.es (J.M.M.); 2Centro de Astrobiología (CAB-INTA), 28850 Torrejón de Ardoz, Spain

**Keywords:** vitamin B_12_, deep subsurface, extreme environments, Iberian Pyrite Belt

## Abstract

Vitamin B_12_ is an enzymatic cofactor that is essential for both eukaryotes and prokaryotes. The development of life in extreme environments depends on cofactors such as vitamin B_12_ as well. The genomes of twelve microorganisms isolated from the deep subsurface of the Iberian Pyrite Belt have been analyzed in search of enzymatic activities that require vitamin B_12_ or are involved in its synthesis and import. Results have revealed that vitamin B_12_ is needed by these microorganisms for several essential enzymes such as ribonucleotide reductase, methionine synthase and epoxyqueosine reductase. Isolate *Desulfosporosinus* sp. DEEP is the only analyzed genome that holds a set core of proteins that could lead to the production of vitamin B_12_. The rest are dependent on obtaining it from the subsurface oligotrophic environment in which they grow. Sought proteins involved in the import of vitamin B_12_ are not widespread in the sample. The dependence found in the genomes of these microorganisms is supported by the production of vitamin B_12_ by microorganisms such as *Desulfosporosinus* sp. DEEP, showing that the operation of deep subsurface biogeochemical cycles is dependent on cofactors such as vitamin B_12_.

## 1. Introduction

The deep subsurface is a poly-extreme environment inhabited by microorganisms, [1,2] where they have to withstand several extreme factors, such as high pressure, low nutrient concentration, high temperatures and low water activity [3]. Additionally, and in contrast with extreme environments on the surface, biogeochemical cycles operate without photosynthesis as a source of organic matter [2]. Thus, biogeochemical cycles depend on chemosynthetic organisms as primary producers, geochemically and biologically produced H_2_ [4,5] and the presence of electron acceptors for anaerobic respiration due to the existing anoxic conditions. To colonize these very extreme niches, microorganisms must cooperate in order to be able to use the few available resources to thrive [1,4,6]. The deep subsurface of the Iberian Pyrite Belt (IPB) has been thoroughly studied via several drilling projects [7,8,9], which have revealed the existence of an underground bioreactor. This bioreactor employs inorganic compounds such as Fe^3+^ to dissolve metal sulfides such as pyrite (FeS_2_) and increase nutrient availability in the deep subsurface. Nitrogen, carbon, iron, hydrogen and sulfur biogeochemical cycles have been confirmed throughout 600 m of rock [10]. These studies also allowed the isolation and sequencing of several microorganisms native to the subsurface and the study of their genomes and metabolic capabilities in silico [11,12,13,14,15] and in vitro [10,11].

Enzymes are often in need of cofactors such as vitamin B12 (Figure 1) that allow chemical reactions to flow. Although these are needed in very small quantities, they are essential for the organism. Cobalamin, also known as vitamin B_12_, is an essential cofactor for life, involved in processes such as DNA methylation and methionine or ribonucleotide synthesis. Vitamin B_12_ is a tetrapyrrole with a covalently linked cobalt atom and belongs to the cobamide group, also known as corrinoids. Additionally, these molecules have an upper ligand and a nucleotide loop with a lower ligand. These two positions differ between the different types of cobamides, and vitamin B_12_ is characterized by a 5,6-dimethylbenzimidazole (DMB) as the lower ligand. However, vitamin B_12_ can have different upper ligands, such as a methyl group (methylcobalamin), a hydroxyl group (hydroxocobalamin), a cyano group (cyanocobalamin) or a 5′-deoxyadenosyl molecule (adenosylcobalamin; Figure 1). In addition, its synthesis is only feasible in a limited variety of archaea and bacteria. As such, both prokaryotes and eukaryotes are heavily dependent on its synthesis by microbial producers for their survival [16]. Prokaryotes produce B_12_ with either a methyl group or with 5′-desoxyadenosyl, while the hydroxyl group is present when the natural forms are exposed to light and the cyano group in synthetic form is produced as a pharmaceutical complement [17].

The complete pathway for the synthesis of vitamin B_12_ ranges between 21 and 30 different reactions, making it a very demanding pathway in terms of energy and evolutionary cost. Its synthesis can take place in the presence or in the absence of oxygen [18]. Therefore, most microorganisms depend on the retrieval of this cofactor from the medium for survival [19]. This is known as the salvage pathway, and it requires a significantly smaller number of proteins and allows microorganisms that cannot synthesize vitamin B_12_ to capture intermediaries from the extracellular medium and modify them into adenosylcobalamin [17]. 

To the best of our knowledge, the importance of vitamin B_12_ in the deep subsurface has not yet been described. The purpose of this study is to analyze 12 genomes which had been previously sequenced to study the ability of these microorganisms to synthesize vitamin B_12_ and to try to infer from this data the relevance of vitamin B_12_ in the deep subsurface of the IPB.

## 2. Materials and Methods

### 2.1. Genomes for In Silico Analysis 

The genomes used in this study belong to twelve microorganisms isolated from the deep subsurface of the IPB (Table 1). These microorganisms were isolated using selective anaerobic media with rocks from the IPB subsurface as inoculum. The taxonomic identification of each isolate was achieved through the sequencing of the 16S rRNA gene [10,20]. Depth detection of each isolate is displayed in Appendix A. Isolation and subsequent growth of isolates were performed in an anaerobic atmosphere and in the absence of light. 

### 2.2. Detection of Cobamide Biosynthesis and Dependence Genes in Genomes

To identify proteins and enzymes which are involved in each step of cobamide biosynthesis or that are cobamide-dependent or independent (Figure 2), we first built a database with several amino acid sequences for each of these proteins retrieved from UniProt (Appendix A). Then we queried (blastp) this database against the proteins predicted for the twelve aforementioned microorganisms (Table 1). We considered as positive those results which had a percent identity higher than 35%, a query cover higher than 50% and an E-value smaller than 10^−5^. Genes that had either the query cover or the percent identity above the threshold but did not meet the requirement on the other were compared with blastp against the UniProt Reference proteomes + Swiss-Prot database with the UniProt Blast suite [21].

### 2.3. Domain Detection and Structure Comparison between BtuB Transporters

Evaluated proteins are the putative BtuB transporters detected in this article and BtuB annotated proteins from *Shewanella putrefaciens* T2.3D-1.1 For the predictions of domains in each protein, we used the InterPro tool [22] with standard parameters (Appendix A). For the prediction of the structure of the putative BtuB transporters we used the algorithm AlphaFold 2 available inside the Galaxy European server [23] with standard parameters and the full database. The predicted structures were compared to the crystal structure of the BtuB transporter 1NQE and to the crystal structure of the FecA transporter (1KMO) from the PDB database [24]. Pairwise alignment was made through the FATCAT (Flexible structure AlignmenT by Chaining Aligned fragment pairs allowing Twists) online server using the flexible alignment mode with standard parameters [25]. 

**Table 1 genes-14-01339-t001:** Microorganisms isolated from the deep subsurface of the IPB with the accession number of their genomes. ENA: European Nucleotide Archive. JGI-IMG: Joint Genome Institute—Integrated Microbial Genomes.

Isolate	Accession Number
*Aestuariimicrobium* sp. T2.26MG-19.2	ERS14911041 (ENA)
*Bacillus* sp. T2.9-1	ERS14952636 (ENA)
*Brevundimonas* sp. T2.26MG-97	UXHF00000000 (ENA)
*Ciceribacter* sp. T2.26MG-112.2 (previously described as *Rhizobium* sp. T2.26MG-112.2)	ERR2572853 (ENA)
*Ciceribacter* sp. T2.30D-1.1 (previously described as *Rhizobium* sp. T2.30D-1.1)	UEYP00000000 (ENA)
*Desulfosporosinus* sp. DEEP	2721755100 (JGI-IMG)
*Microbacterium* sp. T2.11-28	ERS14952639 (ENA)
*Paenibacillus* sp. T2.5-46A	ERS6409820 (ENA)
*Rhodoplanes* sp. T2.26MG-98	UWOC00000000 (ENA)
*Shewanella putrefaciens* T2.3D-1.1	CACVBT020000000 (ENA)
*Stutzerimonas* sp. T2.31D-1 (previously described as *Pseudomonas* sp. T2.31D-1)	CAJFAG010000000 (ENA)
*Tessaracoccus* sp. T2.5-30	CP019229 (ENA)

## 3. Results

### 3.1. B_12_ Dependence in the IPB Subsurface

Enzymes that have been reported to depend on vitamin B_12_ to process their substrates are compared against the database that contains all the proteins from the 12 isolates from the IPB. In similar fashion, those vitamin B_12_-dependent proteins that have vitamin B_12_-independent counterparts or alternative enzymes are also compared to our database (Table 2). In this way we seek to determine which microorganisms are auxotrophs for cobalamin. Expanded results for proteins dependent on B_12_ can be found in Appendix A. Additionally, expanded results for B_12_-independent counterparts proteins can be found in Appendix A.

The vitamin B_12_-dependent methionine synthase MetH (Figure 2A) is present in 11 out of the 12 isolates under study. Nevertheless, 6 out of the 12 also hold a vitamin B_12_-independent methionine synthase MetE (Figure 2B). Only *Desulfosporosinus* sp. DEEP has not returned a match for this enzyme. Hence, *Aestuariimicrobium* sp. T2.26MG-19.2, *Ciceribacter sp.* T2.26MG-112.2, *Ciceribacter* sp. T2.30D-1.1, *Rhodoplanes* sp. T2.26MG-98 and *Tessaracoccus* sp. T2.5-30 would be dependent on vitamin B_12_ for the synthesis of methionine, an essential amino acid. 

In the case of the vitamin B_12_-dependent ribonucleotide reductase NrdJ (Figure 2A), 10 out of the 12 isolates have an annotated copy in their genomes (Table 2). We also searched for B_12_-independent ribonucleotide reductase NrdAB enzymes (Figure 2B) annotated in their genomes and found that 8 out of the 12 isolates had this enzyme (Table 2).

The vitamin B_12_-dependent methylmalonyl-CoA mutase BhbA enzyme (Figure 2A) is also present in 6 out of the 12 genomes (Table 2). To the best of our knowledge there is no vitamin B_12_-independent ortholog for this enzyme.

The vitamin B_12_-dependent epoxyqueosine reductase QueG (Figure 2A) was found in 8 of our isolates (Table 2). Additionally, we have not obtained positive results for the vitamin B_12_-independent epoxyqueosine reductase QueH (Figure 2B; Table 2). 

The vitamin B_12_-dependent glutamate mutase GlmS (Figure 2A) was only found in *Desulfosporosinus* sp. DEEP (Table 2). To the best of our knowledge, there is no vitamin B_12_-independent glutamate mutase, but there are several enzymes involved in the degradation of glutamate that can also use glutamate as substrate. We have looked for two different B_12_-independent proteins that can use glutamate as a substrate, GdhA and AspC. The glutamate dehydrogenase GdhA (Figure 2B) has been detected in all of the genomes except for those of *Desulfosporosinus* sp. DEEP and *Microbacterium* sp. T2.11-28 (Table 2). The aspartate transaminase AspC (Figure 2B) is common to all the assayed genomes except for that of *Aestuariimicrobium* sp. T2.26MG-19.2 (Table 2). Therefore, none of our isolates would depend entirely on vitamin B_12_ for the degradation of glutamate.

Another vitamin B_12_-dependent enzyme that has returned a positive match in 3 isolates (Table 2) is the ethanolamine ammonia lyase EutBC (Figure 2A). There is an alternative pathway for ethanolamine consumption that is not dependent on vitamin B_12_, composed of 7 different genes (Csal_0675 to Csal_0681) [26]. The two isolates from the genus *Ciceribacter* present the complete set of proteins involved in the utilization of ethanolamine without the need for vitamin B_12_ (Table 2). For AcsCDE, a vitamin B_12_-dependent methyltransferase (Figure 2A), we have not found a B_12_-independent counterpart. Nevertheless, this enzyme is present in 4 of the isolate’s genomes (Table 2). 

The last vitamin B_12_-dependent enzyme to return a positive match is BchE (Figure 2A). It is involved in the synthesis of chlorophyll and bacteriochlorophyll, which only returned a positive match in *Rhodoplanes* sp. T2.26MG-98 (Table 2). The vitamin B_12_-independent counterpart AcsF (Figure 2B) also returned a positive match in *Rhodoplanes* sp. T2.26MG-98 (Table 2).

From our set of B_12_-independent proteins, 5 have not returned positive matches in any of the genomes. These include the 2-hydroxybutanoil-CoA mutase HcmB, the 2-methyleneglutarate mutase Mgm, the glycerol/diol dehydratase PduCDE, the reductive dehalogenase PceA and the mercury methyltransferase HgcAB (Table 2; Figure 2A).

### 3.2. B_12_ Production in the IPB Subsurface

According to the last section, there are several subsurface isolates that depend on vitamin B_12_ for very important enzymatic reactions. We looked for the complete array of proteins needed for the synthesis of adenosylcobalamin [16,17] across all our isolates (Appendix A). The first step involves the synthesis of uroporphyrinogen III starting from glycine and succinyl-CoA or glutamate (Figure 2C). From the sample, 4 genomes have all the necessary proteins for the synthesis of this intermediary from either glutamate or glycine and succinyl-CoA. While *Ciceribacter* sp. T2.30D-1.1 has the necessary proteins to start from succinyl-CoA and glycine, it lacks those needed to start from glutamate (Appendix A). 

For the aerobic/anaerobic pathways for the synthesis of adenosylcobyrinic acid (Figure 2C), our results have not found any of the genomes to contain all the proteins ascribed to either of the options (Appendix A). The best results in the aerobic pathway are 10 proteins out of the total 12 proteins for *Tessaracoccus* sp. T2.5-30. The maximum detected for the anaerobic pathway is 10 of the 11 proteins of the anaerobic pathway, for both *Desulfosporosinus* sp. DEEP and *Tessaracoccus* sp. T2.5-30. Even so, a recurring pattern that emerged across this study is that protein IDs from our genomes would appear in both the aerobic and anaerobic pathways for different proteins, hence WP_154722398.1 from *Ciceribacter* sp. T2.26MG-112.2 appears as CobA and CysG (Appendix A). As a result, we have been unable to map some protein IDs to enzymes involved in either the aerobic or anaerobic pathways of synthesis of adenosylcobrynic acid. 

As for the central pathway, *Stutzerimonas* sp. T2.31D-1 is the only isolate to have returned positive results from all the involved proteins (Appendix A). 

Since most of our sampled genomes lack many of the proteins for the synthesis of vitamin B_12_, it was deemed important to compare our genomes against known importers for the salvage of vitamin B_12_. The uptake system of BtuBCDF to salvage vitamin B_12_ is only complete in the *S. putrefaciens* T2.3D-1.1 genome. Aside from the BtuCDF and BtuB system, we also looked for alternative vitamin B_12_ importers BtuM [27], Rv1819c [28] and CbrT [29]. BtuM and Rv1819c returned positive matches in *Stutzerimonas* sp. T2.31D-1 and *Rhodoplanes* sp. T2.26MG-98 respectively. On the other hand, CbrT was not found in any of the assayed genomes (Appendix A).

Although it has been stated previously that the *S. putrefaciens* T2.3D-1.1 genome contains 13 putative BtuB transporters [15], in this work, just 2 different BtuB transporters were detected (CAD6364477.1 and CAD6365988.1). The domain structure of these 13 putative BtuB copies and putative BtuB proteins found in this work (CAD6364477.1, CAD6365988.1, WP_048328632.1 and WP_008262044.1) reveal domains characteristic of TonB-dependent receptors (IPR012910, IPR037066, IPR000531, IPR036942, IPR039426, IPR010916, IPR010917 and IPR010101) (Appendix A). Domains IPR012910, IPR037066, IPR000531, and IPR036942 are shared among 16 out of the 17 assayed proteins. For the remaining domains, IPR039426 returned a positive match in 11 proteins, IPR010917 was detected on 6 proteins, and IPR010916 and IPR010101 are only present in 1 protein. Protein CAD6364477.1 from *S. putrefaciens* T2.3D-1.1 shows the highest number of domains, missing only IPR039426. As such, none of our 17 putative BtuB proteins has the same totality of domains (Appendix A) as reference BtuB’s CAD6020855.1 and RIH46231.1. 

To better understand similarities and differences between annotated BtuB proteins and BtuB proteins found in this article, we predicted the structure of the 13 BtuB annotated copies of *S. putrefaciens* T2.3D-1.1 [15], plus all the BtuB proteins that were been found in this work (Appendix A). Using the predicted structures, we have compared them through pairwise alignment with the *Escherichia coli* BtuB structure from the PDB (1NQE) and with another TonB-dependent receptor (1KMO) as a control (Appendix A). BtuB proteins from this work (WP_048328632.1, WP_008262044.1, CAD6365988.1 and CAD6364477.1) have low *p*-values, attributed to significantly similar proteins, and high FATCAT scores and no twists when compared to 1NQE. When compared to 1KMO the FATCAT scores are lower but are also considered significant similar proteins. Out of these proteins, CAD6365988.1 from *S. putrefaciens* T2.3D-1.1 and WP_008262044.1 from *Brevundimonas* sp. T2.26MG-9 hold the best results with 1NQE (Figure 3) and their comparison with the control 1KMO yielded lower scores (Appendix A). Even though some of the 13 annotated proteins from *S. putrefaciens* T2.3D-1.1 (CAD6367072.1 and CAD6364935.1) have high FATCAT scores with 1NQE, they still have lower scores than the BtuB proteins found in this work. Additionally, some of these annotated proteins, such as CAD6364935.1 and CAD6366340.1, show higher FATCAT scores when compared with the control 1KMO such as CAD6366340.1 and CAD6364935.1 (Appendix A).

## 4. Discussion

In our array of genomes, we have found 13 enzymes dependent on vitamin B_12_ (Table 2) with different degrees of importance. Of these, 8 have returned positive matches in some microorganisms from our sample (Table 2). These are the methionine synthase, glutamate mutase ribonucleotide reductase, epoxyqueosine reductase, methylmalonyl-CoA mutase, ethanolamine ammonia lyase, methyltransferase (anaerobic magnesium-protoporphyrin IX monomethyl ester cyclase) and the bacteriochlorophyll cyclase (Figure 2A). The methionine synthase, methylmalonyl-CoA mutase, epoxyqueosine reductase and ribonucleotide reductase are highly conserved among bacteria [16]. 

The methylmalonyl-CoA mutase enzyme catalyzes the reversible conversion of (R)-methylmalonyl-CoA to succinyl-CoA [30]. It is a very important reaction involved in many pathways such as the fermentation of lactate to propionate, H_2_ and acetate; for the conversion of succinate to propanoate; or in the 3-hydroxypropanotae cycle for the fixation of CO_2_ [31]. The epoxyqueosine reductase is responsible for the production of queosine from epoxyqueosine, which is a modified base present in the wobble position for the synthesis of amino acids such as histidine, aspartic acid, asparagine and tyrosine [32]. The ribonucleotide reductase synthesizes the deoxyribonucleotides essential for DNA synthesis [33], and the methionine synthase is responsible for the synthesis of the essential amino acid methionine [34]. Although conserved, most of these deeply rooted enzymes have vitamin B_12_-independent counterparts, except for the methylmalonyl-CoA mutase. Even so, our isolates depend on vitamin B_12_ for at least one of these four conserved reactions. In particular, *Ciceribacter* sp. T2.26MG-112.2 and *Ciceribacter* sp. T2.30D-1.1 depend on vitamin B_12_ for three highly conserved enzymatic activities among bacteria: methionine synthase, ribonucleotide reductase and the epoxyqueosine reductase (Table 2).

Aside from highly conserved enzymes such as methionine synthase, methylmalonyl-CoA mutase, epoxyqueosine reductase and ribonucleotide reductase, we have also looked for enzymes that are less conserved in bacteria (Table 2). To start with, the vitamin B_12_-dependent glutamate mutase allows the conversion of L-glutamate to L-threo-3-methylaspartate and is involved in the degradation of glutamate [35]. The vitamin B_12_-independent enzymes glutamate dehydrogenase and aspartate transaminase convert L-glutamate into 2-oxoglutarate [36] and catalyze a reversible reaction from L-glutamate to L-aspartate, respectively [37]. Only *Desulfosporosinus* sp. DEEP has returned a positive match against the vitamin B_12_-dependent glutamate mutase, but all the isolates do have at least one vitamin B_12_-independent pathway for glutamate degradation. 

The ethanolamine lyase catalyzes the degradation of ethanolamine to acetaldehyde and ammonia [38]. This nitrogenated compound can be used by some bacteria as a source of N and C. Acetaldehyde is then transformed by other enzymes into acetate and released to the extracellular medium [39]. Alternatively, there is a vitamin B_12_-independent pathway in *Chromobacter salexigens* that would yield glycine instead of acetate similar to the vitamin B_12_-dependent pathway [26]. Thus, for *Stutzerimonas* sp. T2.31D-1, *Tessaracoccus* sp. T2.5-30 and *Desulfosporosinus* sp. DEEP, the final product from ethanolamine consumption would be acetate and ammonia, and for both *Ciceribacter* strains it would be glycine. Nevertheless, NH_4_^+^ and acetate have been detected in the subsurface of the IPB [10]; hence, its abundance could be tied to the production of B_12_. 

The methyltransferase AcsCDE catalyzes the transfer of a methyl group from methyl-tetrahydrofolate to the vitamin B_12_ cofactor of the protein in the fixation of CO_2_ in the Wood–Ljungdahl pathway [40]. Both *Ciceribacter* strains and *Microbacterium* sp. T2.11-28 have returned positive matches for at least one of the three subunits, and only *Desulfosporosinus* sp. DEEP has matched the three subunits of this transferase (Appendix A). Different strains of the *Desulfosporosinus* genus have been reported to fixate CO_2_ through the reductive acetyl-CoA pathway [41,42,43], and our isolate does possess all the necessary proteins except for FdhB [10]. Thus, *Desulfosporosinus* sp. DEEP is a good candidate for CO_2_ fixation in the subsurface of the IPB [10].

The bacteriochlorophyll cyclase enzyme, or BchE, requires vitamin B_12_ for the synthesis of 3,8-divinylprotochlorophyllide, an intermediary in the formation of the isocyclic ring of chlorophylls [44]. Its B_12_-independent counterpart, AcsF, can act under both aerobic and anaerobic conditions [45]. Due to the anoxic conditions and the absence of light in the subsurface of the IPB, it is as yet unknown what role the synthesis of chlorophylls could have. Regardless, this is not the first report of microorganisms that are deemed photosynthetic living in the continental subsurface [6,10]. In our sample, only *Rhodoplanes* sp. T2.26MG-98 returned positive results for both the vitamin B_12_-dependent and the vitamin B_12_-independent enzymes. Although not entirely dependent, vitamin B_12_ would still be needed for the synthesis of chlorophyll. 

Of the analyzed genomes, *Ciceribacter* sp. T2.26MG-112.2 and *Ciceribacter* sp. T2.30D-1.1 would depend on vitamin B_12_ for three highly conserved enzymatic activities among bacteria, namely methionine synthase, ribonucleotide reductase and the epoxyqueosine reductase. Other isolates depend on vitamin B_12_ for less conserved reactions, such as *Stutzerimonas* sp. T2.31D-1 and *Tessaracoccus* sp. T2.5-30 for the consumption of ethanolamine or the fixation of CO_2_ in the case of *Desulfosporosinus* sp. DEEP. Overall, all our isolates depend on vitamin B_12_ in at least one of the cobalamin-dependent activities that have been studied. These results shed light on the requirements for life in the deep subsurface. Vitamin B_12_ may play a relevant role as a cofactor and as a key component to the development of its communities and for some of its metabolisms. 

Considering the need for vitamin B_12_ for the given set of very ubiquitous microorganisms of the IPB [10], we have also looked for the set of proteins required for the synthesis of vitamin B_12_. The synthesis of vitamin B_12_ involves a very complex and extensive list of proteins (Figure 2C) [17]. The first step involves the synthesis of the precursor uroporphyrinogen III, which is shared among different tetrapyrrole-based molecules such as the heme group, chlorophylls or the coenzyme F430 [18]. There are two different starting points; one depends on glutamate and uses the proteins GtlX and HemAL to produce 5-aminolevulinic acid, and the other path starts from glycine and succinyl-CoA to also yield 5-aminolevulinic catalyzed by HemA. Afterwards, the proteins HemBCD produces uroporphyrinogen III in both cases. For the construction of the coring ring, uroporphyrinogen III can then be processed through an aerobic or anaerobic pathway. Shelton and colleagues (2018) [19] conducted a larger analysis of the production of vitamin B_12_ and obtained similar results, reporting that some proteins involved in the aerobic/anaerobic pathways CobI:CbiL, CobJ:CbiH, CobM:CbiF, CobK:CbiJ, CobL:CbiT, CobL:CbiE, CobM:CbiC and CobB:CbiA are orthologous (Figure 2C). Therefore, differentiating between aerobic and anaerobic pathways based on the sequence alone was untenable. Finally, the central pathway, where the nucleotide loop is assembled, is shared between the aerobic and anaerobic pathways and produces adenosylcobalamin (vitamin B_12_) from adenosylcobyrinic acid [18].

All in all, although none of our genomes have shown a complete vitamin B_12_ biosynthetic pathway, *Desulfosporosinus* sp. DEEP, *Ciceribacter* sp. T2.26MG-112.2, *Ciceribacter* sp. T2.30D-1.1, *Rhodoplanes* sp. T2.26MG-98 and *Tessaracoccus* sp. T2.5-30 have most of them (Appendix A). Additionally, Shelton and colleagues (2018) [19] found a core set of eight proteins for the synthesis of this cofactor that are shared between the genomes of in vitro tested vitamin B_12_ producers (Table 2). According to this criterion, only *Desulfosporosinus* sp. DEEP holds the core set of genes necessary to synthesize vitamin B_12_. Therefore, *Desulfosporosinus* sp. DEEP could play a central role in the ecosystem of the IPB not only as a vitamin B_12_ producer but possibly as a primary producer as well through the fixation of CO_2_ via the Wood–Ljungdahl pathway. Nevertheless, *Ciceribacter* sp. T2.26MG-112.2, *Ciceribacter* sp. T2.30D-1.1, *Rhodoplanes* sp. T2.26MG-98 and *Tessaracoccus* sp. T2.5-30 contain all the proteins except CobC, making them good candidates for vitamin B_12_ production as well. 

Even so, intermediaries of the vitamin B_12_ biosynthesis or the vitamin itself can also be obtained from the extracellular medium and constitute a very relevant interaction between producers and vitamin B_12_ auxotrophs at different ecosystemic levels [46,47,48]. The proteins involved in the salvage of vitamin B_12_ belong to the central pathway (CobU/CobP, CobS/CobV and CobC) and different transporters for its uptake [17]. The most studied protein complex for the capture of vitamin B_12_ is the ABC system BtuCDF with the outer membrane receptor BtuB [49,50]. With the exception of *S. putrefaciens* T2.3D-1.1, none of our isolates, although dependent, would have the means to capture extracellular vitamin B_12_ through this system. Reference BtuB proteins CAD6020855.1 and RIH46231.1 have 7 different domains (the plug domain (IPR012910 and IPR037066), the interaction with TonB and β-barrel domains (IPR000531, IPR036942), the TonB Box (IPR010916), the TonB-dependent receptor conserved site (IPR010917 and IPR039426) and the BtuB specific domain (IPR010101)). When compared with the annotated BtuB transporters from *S. putrefaciens* T2.3D-1.1 on the domain level, only CAD6364477.1 from *S. putrefaciens* T2.3D-1.1 shows 6 out of the 7 domains (Appendix A). Since the BtuB proteins found in *S. putrefaciens* T2.3D-1.1 in this work were different from those annotated previously [11], a more in-depth study was performed to reinforce the obtained results.

Structure prediction and comparison with the model 1NQE confirms that our putative BtuB are closer to BtuB than the control structure 1KMO. The control structure 1KMO belongs to the outer membrane transporter FecA [51], which was also a result in the Blastp search for BtuB transporters. We used 1KMO as a control TonB-dependent receptor to see if we could discern it from 1NQE using both structure and sequence to study putative TonB-dependent transporters. Structural differences between TonB-dependent transporters, such as FecA and BtuB, are very small, and substrate specificity depends on the outer loops between β-barrell strands [49], accounting for a low number of amino acids out of the total sequence that conform the protein. Therefore, the information that can be obtained from these structures is limited. When looking for BtuCDFB proteins, most of the results are ABC transporters involved in the transport of siderophores, hemin, sulphate/thiosulphate, and, to a lesser extent, cobalamin. Hence, we have found many ABC transporters in the genomes of our isolates, but we remain unable to assign specific functions to many of these transporters based only on their sequence. As stated above, based only on sequence, it is very difficult to ascertain what type of ligands and the specificity of such transporters might be. Nevertheless, the combination of sequence-based screening and protein structure prediction shows that the BtuB transporters detected in this work resemble the reference model more closely than the annotated vitamin B_12_ transporters from *S. putrefaciens* T2.3D-1.1. 

Aside from the BtuCDF and BtuB system, we have also looked for alternative vitamin B_12_ importers. BtuM is a cobalamin transporter that can transport vitamin B_12_ to the periplasm in cooperation with BtuB without the need for BtuCDF. Additionally, BtuM can also remove the CN^−^ group from cyanocobalamin, allowing its conversion to physiological forms [27]. BtuM has only been found in *Stutzerimonas* sp. T2.31D-1. It also has been suggested that proteins such as BtuR could further transport vitamin B_12_ in collaboration with BtuM [51], which is more widespread in our sample and is present in *Stutzerimonas* sp. T2.31D-1 as well. Another alternative is Rv1819c, which has been described as a hydrophilic compound transporter that could also transport vitamin B_12_ [28] and has been detected in the *Rhodoplanes* sp. T2.26MG-98 genome. Therefore, *Stutzerimonas* sp. T2.31D-1 and *Rhodoplanes* sp. T2.26MG-98 have other alternatives in obtaining vitamin B_12_. 

Based on our results, all isolates depend on vitamin B_12_ for essential enzymatic reactions such as the methionine synthase in the case of *Aestuariimicrobium* sp. T2.26MG-19.2, *Ciceribacter* sp. T2.26MG-112.2, *Ciceribacter* sp. T2.30D-1.1, *Rhodoplanes* sp. T2.26MG-98 and *Tessaracoccus* sp. T2.5-30 or the ribonucleotide reductase for *Ciceribacter* sp. T2.26MG-112.2, *Ciceribacter* sp. T2.30D-1.1, *Rhodoplanes* sp. T2.26MG-98 and *Tessaracoccus* sp. T2.5-30 (Table 2). Despite being unable to find the putative vitamin B_12_ transporter for many of our isolates, other ways to introduce this cofactor to the cytoplasm must exist to ensure its development in the deep subsurface.

In this study, we conclude that some of the isolates depend on vitamin B_12_ to carry out essential functions for their survival. Additionally, some of them, such as *Desulfosporosinus* sp. DEEP, might be able to synthesize vitamin B_12_. These factors have great relevance in the context of the IPB. During the IPBSL project, different techniques were used to identify the microorganisms present along the length the borehole. Moreover, the results of this project allowed the reconstruction of C, N, Fe, H and S biogeochemical cycles that take place in the subsurface of the IPB [10]. Thanks to all this information, we know that the isolates studied in this project are distributed throughout the subsurface and that they play a role in the aforementioned cycles. As a potential vitamin B_12_ producer, *Desulfosporosinus* sp DEEP is distributed throughout the column and has been detected in 14 out of the 43 sampled depths. Particularly, at depths of 416 mbs (meters below surface) and 594 mbs, 8 out of 12 microorgansms have been detected, including *Desulfosporosinus* sp DEEP. Thus vitamin B_12_ sharing seems plausible for most of our sample. This could most certainly enhance the flow of the biogechemical cycles that take place in the deep subsurface of the IPB, meaning that the synthesis of vitamin B_12_ in the subsurface is essential for the operation of the bioreactor. At these depths, we hypothesize that there must be close interaction between these isolates, in which some of them would produce vitamin B_12_ which then would be used by the others. Additionally, microorganisms, such as *S. putrefaciens* T2.3D-1.1 could be sharing other important molecules such as H_2_ [11] as an electron donor, and that has been reported as a substrate for some *Desulfosporosinus* strains [42]. As such, our results would suggest that these microorganisms coexist and develop together in the deep subsurface through cooperation and sharing of essential molecules such as vitamin B_12_.

## 5. Conclusions

In spite of the limitations inherent to the given methodology, we have detected several vitamin B_12_-dependent enzymatic reactions with no vitamin B_12_-independent counterparts in the genomes of our twelve isolates from the deep subsurface. However, we have not been able to ascertain the entry pathways for vitamin B_12_ in all the isolates, so questions regarding this are still open. Although not complete sensu stricto, we have found a complete core set of proteins [24] that could lead to the synthesis of cobalamin in the genome of *Desulfosporosinus* sp. DEEP and an almost complete set in the genomes of *Ciceribacter* sp. T2.30D-1.1, *Ciceribacter* sp. T2.26MG-112.2 and *Rhodoplanes* sp. T2.26MG-98. All in all, we have found 12 isolates with different cobalamin needs, and some of them are promising vitamin B_12_-producing candidates which may possibly meet those needs. The deep subsurface, although vastly unexplored, seems to depend on very similar factors and micronutrients such as vitamin B_12_ when compared with above ground ecosystems. Hence cooperative interactions between microorganisms in poly-extreme environments such as the deep subsurface [10] could be responsible for the colonization and ecopoiesis of the IPB.

## Figures and Tables

**Figure 1 genes-14-01339-f001:**
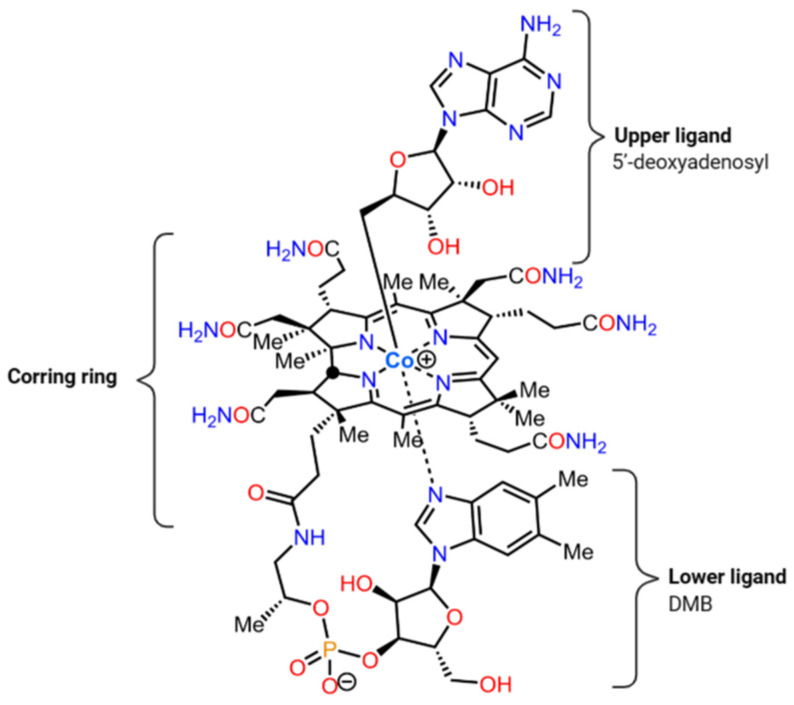
Vitamin B_12_ (adenosylcobalamin) structure with 5′-deoxyadenosil as the upper ligand and 5,6-dimethylbenzimidazole (DMB) as the lower ligand. Modified from Wikipedia (https://en.wikipedia.org/wiki/Adenosylcobalamin), accessed on 23 June 2023.

**Figure 2 genes-14-01339-f002:**
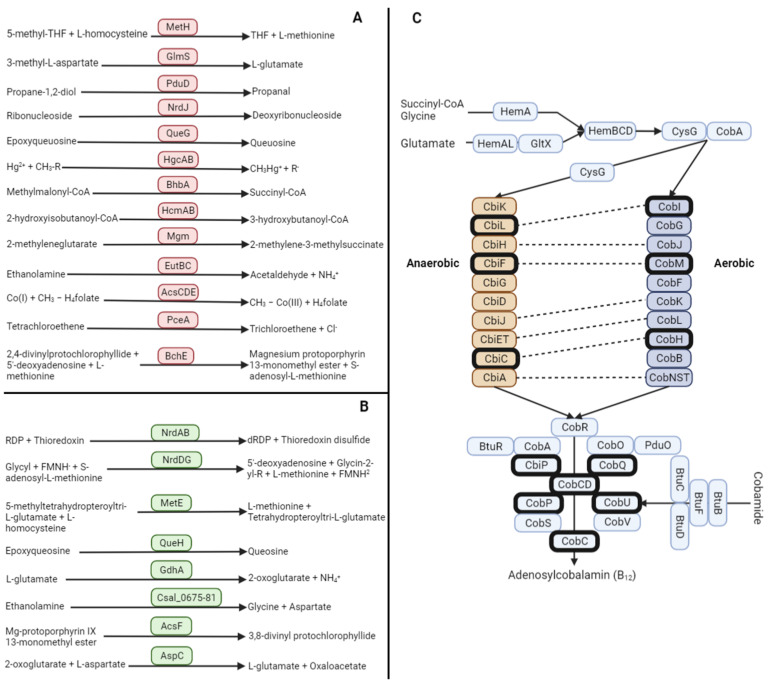
(**A**). Cobalamin-dependent enzymes, enclosed in red squares, with their corresponding reactions. (**B**). Cobalamin-independent counterparts, enclosed in green squares, with their corresponding reactions. (**C**). Adenosylcobalamin synthesis pathway; the anaerobic pathway represented by orange squares, the aerobic pathway by blue squares, and the rest of the proteins by light blue squares. Squares with black borders represent proteins that are essential for the synthesis of B_12_ and the dashed lines indicate which proteins are orthologs based on their sequence [19].

**Figure 3 genes-14-01339-f003:**
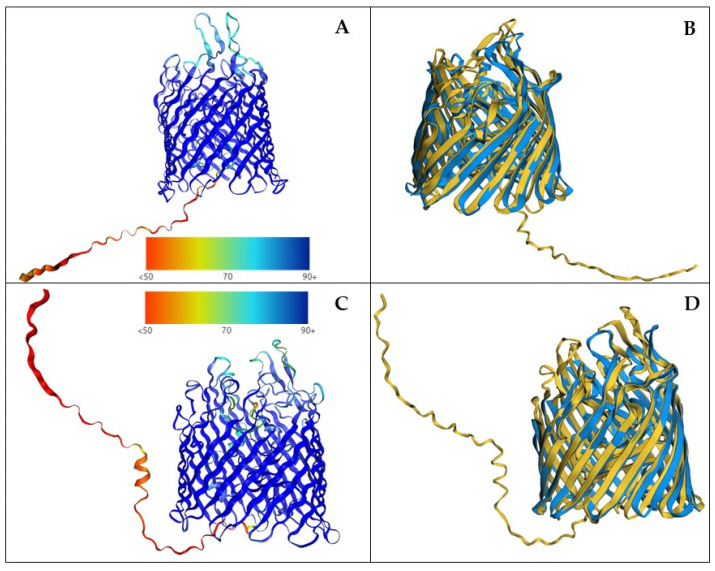
3D structures of proteins WP_008262044.1 and CAD6365988.1 created with alphafold and compared to reference model 1NQE from *E. coli*. Legend for per-residue confidence score (pLDDT) can be found under each alphafold2 predicted structure (**A**). Alphafold2 prediction of WP_008262044.1 structure. (**B**). FatCat pairwise alignment of WP_008262044.1(yellow) and 1NQE (blue) structures. (**C**). Alphafold2 prediction of CAD6365988.1. (**D**). FatCat pairwise alignment of CAD6365988.1 (yellow) and 1NQE Sstructures.

**Table 2 genes-14-01339-t002:** Cobalamin-dependent and-independent counterparts that have been queried against the genomes of the isolates. Green squares indicate positive results, white squares indicate that no positive results have been obtained for the specific protein.

		*Aestuariimicrobium* sp. T2.26MG-19.2	*Bacillus* sp. T2.9-1	*Brevundimonas* sp. T2.26MG-97	*Ciceribacter* sp. T2.26MG-112.2	*Ciceribacter* sp. T2.30D-1.1	*Desulfosporosinus* sp. DEEP	*Microbacterium* sp. T2.11-28	*Paenibacillus* sp. T2.5-46A	*Rhodoplanes* sp. T2.26MG-98	*Shewanella putrefaciens* T2.3D-1.1	*Stutzerimonas* sp. T2.31D-1	*Tessaracoccus* sp. T2.5-30
**Cobamide-dependent enzymes**	**Methionine synthase** **(MetH)**												
	**Glutamate mutase** **(GlmES)**												
	**Ribonucleotide reductase** **(NrdJZ)**												
	**Epoxyqueuosine reductase** **(QueG)**												
	**Methylmalonyl CoA mutase (BhbA, MutA, ScpA)**												
	**2-hydroxyisobutnaoyl-CoA mutase (HcmB)**												
	**2-methyleneglutarate mutase** **(Mgm)**												
	**Ethanolamine ammonia lyase** **(EutBC)**												
	**Glycerol/diol dehydratases** **(PduCDE)**												
	**Methyltransferase** **(AscCDE)**												
	**Reductive dehalogenase** **(PceA)**												
	**Bacteriochlorophyll cuyclase** **(BchE)**												
	**Mercury methyltransferase** **(HgcAB)**												

**Cobamide-independent enzymes**	**Ribonucleotide reductase** **(NrdAB, NrdDG)**												
	**Methionine synthase** **(MetE)**												
	**Epoxyqueuosine reductase** **(QueH)**												
	**Ethanolamine utilization (Csal_0675,0679,0680,0681)**												
	**Chlorophyll biosynthesis** **(AcsF)**												
	**Glutamate degradation** **(GdhA, AspC)**												

## Data Availability

The data presented in this study are available in the article and Supplementary Material.

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
