# Peer review of "Vitamin B12 Auxotrophy in Isolates from the Deep Subsurface of the Iberian Pyrite Belt"

_genes, 2023, doi:10.3390/genes14071339_

Round 1
Reviewer 1 Report
It is a well written manuscript, that can be easily to understand. To round off the story, a few sentences about the isolation strategy and methodology and the species identification could be mentioned
no comments
Author Response
It is a well written manuscript, that can be easily to understand. To round off the story, a few sentences about the isolation strategy and methodology and the species identification could be mentioned
We have expanded further the information regarding isolation, and species identification (lines 99-105). Regardless this was done long before this article was written and as such the proper references were added.
Reviewer 2 Report
The material sent to me does not include supplementary tables. The file name for supplementary tables includes really only the five article tables which are also inside text. This makes the evaluation work more difficult.
As such your tables and figures are clear.
In addition, there are two details which should be improved to make the reading more easy.
The first one is a list of all abbreviations. Make it to the beginning of the paper!
The second issue is that the tables and figures are following each other in part 3.2. Please set them to the place where they are mentioned the first time as guided by the journal.
In tine 77 describe better the isolation and the depths.
The microorganisms isolated might live at least under light – possible under aerobic environments or not? Would you like to comment on this?
You could describe better the cowork between the isolated microorganisms.
Make the reference list according to the guidance given by journal to authors.
Author Response
The material sent to me does not include supplementary tables. The file name for supplementary tables includes really only the five article tables which are also inside text. This makes the evaluation work more difficult.
We are deeply sorry to read that. It appears we made a mistake while uploading the supplementary files, we will contact the editors to change this immediately. Thank you for letting us know.
As such your tables and figures are clear.
In addition, there are two details which should be improved to make the reading more easy.
The first one is a list of all abbreviations. Make it to the beginning of the paper!
Done, we have included a new part in at the beginning of material & methods where we include all the abbreviations that we have used throughout the text (lines 92 to 96)
The second issue is that the tables and figures are following each other in part 3.2. Please set them to the place where they are mentioned the first time as guided by the journal.
We have distributed the tables and figures throughout the text to fit the content better.
In line 77 describe better the isolation and the depths.
We have included more information about the isolation and identification of the isolates in materials & methods (from line 99 to line 105).
The microorganisms isolated might live at least under light – possible under aerobic environments or not? Would you like to comment on this?
You could describe better the cowork between the isolated microorganisms.
We have not tested the growth of this microorganism in photosynthetic conditions, regardless there have been findings of photosynthetic microorganisms like cyanobacteria (see reference 6 “Viable cyanobacteria in the deep continental subsurface”. Perhaps in the right conditions some of our isolates might be able to use light as an energy source or even radiation. Regardless, we have not expanded on this topic in the article because we feel it’s too broad to tackle.
From line 487 to line 502 we have described better the possible interactions between isolates using their depths and possible metabolisms.
Reviewer 3 Report
In the objective of looking for the entry pathway of vitamin B12 in the microorganism in the deep subsurface of the Iberian Pyrite Belt, this study has detected several Vitamin B12-dependent enzymatic reactions. The overall manuscript was well prepared. However, some points should be revised before publication.
- I recommend the author change the title of this paper to fit the content and the limited data obtained in this study.
- The Figures, Table, and Schemes should appear together with the text described. It should not be separated as a part of the Results section.
English language is good
Author Response
- I recommend the author change the title of this paper to fit the content and the limited data obtained in this study.
We have changed the title to “ Vitamin B12 auxotrophy in isolates from the deep subsurface of the Iberian Pyrite Belt”
- The Figures, Table, and Schemes should appear together with the text described. It should not be separated as a part of the Results section.
We have rearranged the tables and figures throughout the text in order to fit the content better.
Round 2
Reviewer 2 Report
Four small corrections:
lines 37-38 write carbon, iron, hydrogen and sulfur!
Empty line before lies 126!
Use italic for all scientific names as line 333 Chromobacter salexigenes and line 630 Chlamydomonas auxotroph
Text is easy to read.
Reviewer 3 Report
The paper was sufficient improved and can be accepted for publication.